# Metal-organic frameworks with photocatalytic bactericidal activity for integrated air cleaning

Ping Li[1], Jiazhen Li[1], Xiao Feng[1], Jie Li[1], Yuchen Hao[1], Jinwei Zhang[1], Hang Wang[1], Anxiang Yin[1], Junwen Zhou[1], Xiaojie Ma[1] & Bo Wang [1]

Air filtration has become an essential need for passive pollution control. However, most of the commercial air purifiers rely on dense fibrous filters, which have good particulate matter (PM) removal capability but poor biocidal effect. Here we present the photocatalytic bactericidal properties of a series of metal-organic frameworks (MOFs) and their potentials in air pollution control and personal protection. Specifically, a zinc-imidazolate MOF (ZIF-8) exhibits almost complete inactivation of *Escherichia coli* (*E. coli*) (>99.9999% inactivation efficiency) in saline within 2 h of simulated solar irradiation. Mechanistic studies indicate that photoelectrons trapped at $Zn^+$ centers within ZIF-8 via ligand to metal charge transfer (LMCT) are responsible for oxygen-reduction related reactive oxygen species (ROS) production, which is the dominant disinfection mechanism. Air filters fabricated from ZIF-8 show remarkable performance for integrated pollution control, with >99.99% photocatalytic killing efficiency against airborne bacteria in 30 min and 97% PM removal. This work may shed light on designing new porous solids with photocatalytic antibiotic capability for public health protection.

---

[1] Beijing Key Laboratory of Photoelectronic/Electrophotonic Conversion Materials, Key Laboratory of Cluster Science, Ministry of Education, School of Chemistry and Chemical Engineering, Beijing Institute of Technology, 100081 Beijing, China. Correspondence and requests for materials should be addressed to X.M. (email: xiaojiema@bit.edu.cn) or to B.W. (email: bowang@bit.edu.cn)

Air is the most critical supply for vast majority of living species on the planet and the quality of which has direct effect on human health. Due to the insufficient air exchange and increased time people spent in confined spaces, such as home, schools, offices, vehicles, and hospitals, the air contaminants are of particular concern, especially in view of their threat to vulnerable groups like children, seniors, and patients[1–4]. Air purifiers and fresh air ventilators are widely used air freshening equipment to regulate indoor air quality and protect people from health hazards like particulate matter (PM), bioaerosols, and volatile organic compounds (VOCs), and so on[5,6]. Most of these air cleaning devices are equipped with fibrous filters, which have the character of dense reticular meshwork and necessary thickness[7,8]. Due to the physical barrier and adhesion effect, these filters show good capture efficiency for PM[6,9], meeting the high efficiency particulate air standard. But significant disadvantages exist for these commonly used fibrous filters, including high air resistance and limited inhibition ability against harmful microorganisms[8,10–12]. In particular, they are only capable of partial retention of pathogens like bacteria, fungi, and virus onto the filter surface, instead of completely cleaning them out[11,12]. Eventually with accumulated organic pollutants as nutrients, the filters become easy breeding ground for pathogenicity microorganism, which causes second airborne contamination[13]. Meanwhile, the accumulation of organism may contribute to the reduction of ventilation volume and loss of filter life. Therefore, it is highly desired to develop integrated filtration materials, which can remove PM effectively as well as interrupt transmission of germs in air by killing them completely.

The coexistence of pathogens with other pollutants (PM and VOCs, etc.) in natural air environment[14,15] makes biological decontamination complicated and challenging. The routine air disinfection techniques, including chemical spraying inactivation and ultraviolet (UV) irradiation inactivation, have some prominent problems limiting their further development and practical application[16–19]. For instance, the extensive use of traditional chemical sanitizing agents (chlorine dioxide, ethylene oxide, etc.) is highly energy consuming and also tends to form harmful byproducts in the presence of other air contaminants[16,20]. UV sterilization can be an useful method[21], yet it lacks of sustained effect, inevitably leads to ozone pollution, and shows low antibacterial efficiency with sunlight as light source. Alternatively, heterogeneous photocatalysis emerges to be an efficient and cost-effective approach to eliminate biological pollution[22–25]. The generated reactive oxygen species (ROS), such as hydroxyl radical ($\cdot$OH), superoxide ($\cdot O_2^-$), singlet oxygen ($^1O_2$), and hydrogen peroxide ($H_2O_2$), can act as strong oxidants to destroy harmful microorganisms[24–27]. Semiconductors like ZnO and $TiO_2$ are potential photocatalysts with biocidal activity and exhibit good performance in air disinfection[24,28]. However, their disinfection efficiency is far from satisfactory, especially under high air flow velocity combined with other contaminants like PM and VOCs.

Metal-organic frameworks (MOFs), an emerging class of porous crystalline materials, have been vigorously investigated in the field of gas storage, separation, and catalysis[29]. Recently, we and others had embarked on studying possibilities of MOFs as adsorbents and catalysts for air pollution control[30–32]. Progress has been made for highly efficient PM removal by developing MOF-based filters (MOFilter) in our previous work[33,34]. Large surface area, high porosity, well-dispersed active centers, and tunable functionalities enable MOFs not only good candidates for air filtration but also promising heterogeneous photocatalysts for air pollutants oxidation[35]. Particularly, MOFs offer us an opportunity to optimize the photocatalytic performance at the molecular level by rationally tuning metal clusters or organic linkers[36,37], which is regarded as a significant competitive advantage of MOFs over traditional semiconductors. Owing to the extraordinary designability of MOFs, successful applications have been realized in photocatalytic area like water splitting[38], $CO_2$ reduction[39], ROS-dominated toxic chemicals oxidation[40], and so on. We decided to explore the possibilities of MOFs for efficient light-catalyzed air sterilization. In combination with strong PM filtration capability and intriguing antimicrobial activity, we strive to develop efficient and integrated air filters based on MOFs (Fig. 1).

Herein, after careful comparison, ZIF-8 (zinc-imidazolate MOF) was selected from five typical water stable MOFs. Superior to ZIF-11 (zinc-benzimidazole MOF) and three representative MOF photocatalysts, including MIL-100(Fe), $NH_2$-MIL-125(Ti), and $NH_2$-UIO-66(Zr), ZIF-8 exhibits >99.9999% inactivation efficiency against *Escherichia coli* (*E. coli*) in saline under 2 h of simulated solar irradiation. Significantly, ZIF-8 also outperforms the extensively used biocidal photocatalysts ZnO and anatase $TiO_2$ in both inactivation rate and efficiency. Electron paramagnetic resonance (EPR) measurement, ROS quenching experiments combined with $Zn^{2+}$ leaching, and germicidal test

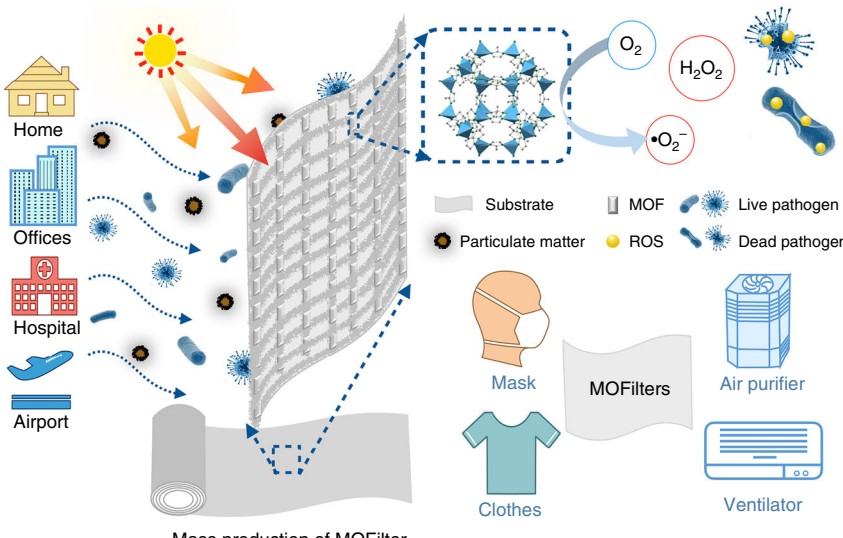

**Fig. 1** Schematic of metal-organic framework (MOF)-based filter. Schematic representation of MOF-based filter (MOFilter) for integrated air cleaning

confirm that $O_2$ could be readily activated by photoinduced $Zn^+$ intermediate catalytic centers within porous MOF to produce $\bullet O_2^-$ via electron transfer, and the formation of ROS like $\bullet O_2^-$ and $H_2O_2$ is responsible for the excellent bactericidal activity of ZIF-8. Benefiting from its favorable feature, ZIF-8 was further processed as the MOFilter showing outstanding performance for integrated air pollution control and personal protection. MOFilter serves as a filtration media with 97% PM capture efficiency achieved at a flow rate of $0.7\,m\,s^{-1}$, as well as a photoinduced contact-killing surface toward airborne bacteria with >99.99% bactericidal efficiency realized in 30 min. This study highlights the importance of MOFs in environmental photocatalysis, and may give new ideas for the design of porous photobactericidal materials.

## Results

**Photocatalytic disinfection performance of ZIF-8.** We initially examined the photocatalytic antibacterial activity of five typical water stable MOFs, constructed from different metal or ligands (MIL-100(Fe), $NH_2$-MIL-125(Ti), $NH_2$-UIO-66(Zr), ZIF-11(Zn), and ZIF-8(Zn)) (Supplementary Figs. 1, 2). The disinfection experiments were conducted in 0.9% (w/v) saline with initial $E.$ $coli$ cell density at $10^7$ colony-forming unit (CFU) $mL^{-1}$ and catalyst dosage at $500\,mg\,L^{-1}$. A 300 W Xe lamp coupled with an AM 1.5 filter ($300\,nm < \lambda < 1100\,nm$) was employed as light source with optical power density fixed at $100\,mW\,cm^{-2}$ (1 sun) to simulate sunlight irradiation. As shown in Fig. 2a, b, ZIF-8 exhibits almost complete inactivation of $E.$ $coli$ (inactivation efficiency of >99.9999%, equivalent to 6.1, $-\log_{10}(C/C_0)$) after 120 min of light irradiation, whereas light alone and ZIF-8 in dark

both show negligible toxic effect on $E.$ $coli$ over the same time period. Particularly, only <77% inactivation efficiency (0.6, $-\log_{10}(C/C_0)$ efficiency) was achieved with light alone. ZIF-8 is greatly superior to MIL-100(Fe), $NH_2$-MIL-125(Ti), and $NH_2$-UIO-66 (Zr) in photoinduced antibacterial performance, which are three representative MOF photocatalysts exhibiting high activity in photocatalytic reactions like $CO_2$ reduction[35,41]. ZIF-8 also shows much stronger photocatalytic inactivation effect on $E.$ $coli$ by comparing with ZIF-11. Encouraged by the excellent photocatalytic characteristic of ZIF-8, we further studied its antibacterial activity in detail. We found that the antibacterial efficiency was largely dependent on the dosage of ZIF-8. We also achieved 99.999% reduction of $E.$ $coli$ by lowering the catalyst dosage to $300\,mg\,L^{-1}$ (Supplementary Fig. 3). Fluorescence assays of $E.$ $coli$ cells with different illuminating time were conducted to confirm the light-induced bacteria-killing behavior of ZIF-8 (Supplementary Fig. 4). The number of dead bacteria increased with irradiation time, and few $E.$ $coli$ cells survived after 120 min. And, there was no bacteria recovery observed in the following 3 days after removing ZIF-8 and light, indicating the irreversible destruction of $E.$ $coli$ (Supplementary Fig. 5). Very importantly, ZIF-8 stays stable in 0.9% (w/v) saline aqueous under 120 min of light illumination (Supplementary Fig. 6). Although trace leached $Zn^{2+}$ ions ($2.65\,mg\,L^{-1}$) were detected by inductively coupled plasma mass spectrometry (ICP-MS), it was proved that a small amount of ZIF-8 precursors including $Zn^{2+}$ and 2-methylimidazole (H-MeIM) had negligible effect on bacteria reduction comparing with blank control ($Zn^{2+}$ ions at $3\,mg\,L^{-1}$ gives 93.5% inactivation efficiency; H-MeIM at $7\,mg\,L^{-1}$ gives 73.7% efficiency; blank control: light irradiation without any catalyst gives 76.3% inactivation efficiency under the identical

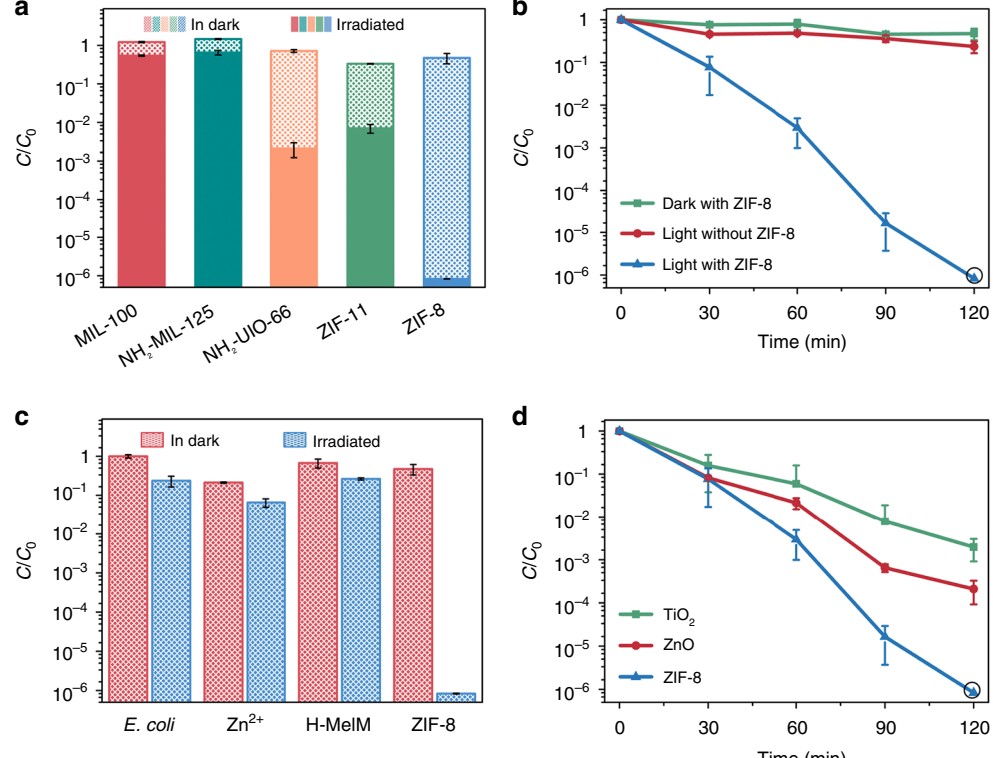

**Fig. 2** Photocatalytic disinfection performance of ZIF-8 (zinc-imidazolate MOF). **a** Disinfection performance comparison among five metal-organic frameworks (MOFs). **b** Inactivation kinetics of $E.$ $coli$ in the presence of ZIF-8. **c** Inactivation efficiency against $E.$ $coli$ in the presence of $Zn^{2+}$ ($3\,mg\,L^{-1}$), 2-methylimidazole (H-MeIM) ($7\,mg\,L^{-1}$), and ZIF-8 ($500\,mg\,L^{-1}$), respectively. **d** Disinfection performance comparison among ZIF-8, $TiO_2$, and ZnO under simulated solar irradiation. In the disinfection performance, the error bars are calculated via repeating the measurements for three times and the black circle represents no measurable levels of bacteria in the culture medium

conditions) (Fig. 2c)[42,43]. It is noteworthy that the levels of $Zn^{2+}$ released from ZIF-8 are much lower than the minimum inhibitory concentration (MIC, 31.25 mg L$^{-1}$) and minimum bactericidal concentration (MBC, 250 mg L$^{-1}$) (Supplementary Fig. 7 and Supplementary Table 1). This indicates that the remarkable antibacterial effect arises from the intact ZIF-8 in the presence of light irradiation. Moreover, the photocatalytic disinfection activity of ZIF-8 was evaluated by comparing with two traditional semiconductive photocatalysts, anatase $TiO_2$ and ZnO (Supplementary Fig. 8). We can see that ZIF-8 is much more active than the above two materials. Over 120 min the log inactivation efficiency of ZIF-8 (6 log) is three and four times higher than that of ZnO (3 log) and anatase $TiO_2$ (2 log), respectively[28,44] (Fig. 2d). Also, the first-order bactericidal rate of ZIF-8 (0.05 min$^{-1}$) is faster than that of contrasting materials (0.02 and 0.03 min$^{-1}$ for anatase $TiO_2$ and ZnO, respectively) (Supplementary Fig. 9). This suggests that nanoporous structure is beneficial for the

enhancement of photocatalytic activity, due to the possible shortened migration distance of photoinduced carriers from bulk to catalyst surface[41].

**Band-structure characterization of ZIF-8.** Given the satisfactory disinfection performance of ZIF-8, its photocatalytic mechanism was further studied. First, the semiconductor properties of ZIF-8 were characterized, since the band structure can directly affect the photocatalytic disinfection efficiency and mechanism. ZIF-8 shows two absorption peaks at 227 and 350 nm, respectively (Supplementary Fig. 10). The strong absorption centered at 227 nm could be attributed to intraligand charge transfer, and shows a slight red shift relative to H-MeIM. The broad absorption at longer wavelength of 350 nm is responsible for ligand to metal charge transfer (LMCT), as confirmed by the low-temperature EPR study (Fig. 3b). Accordingly, the bandgap energy was

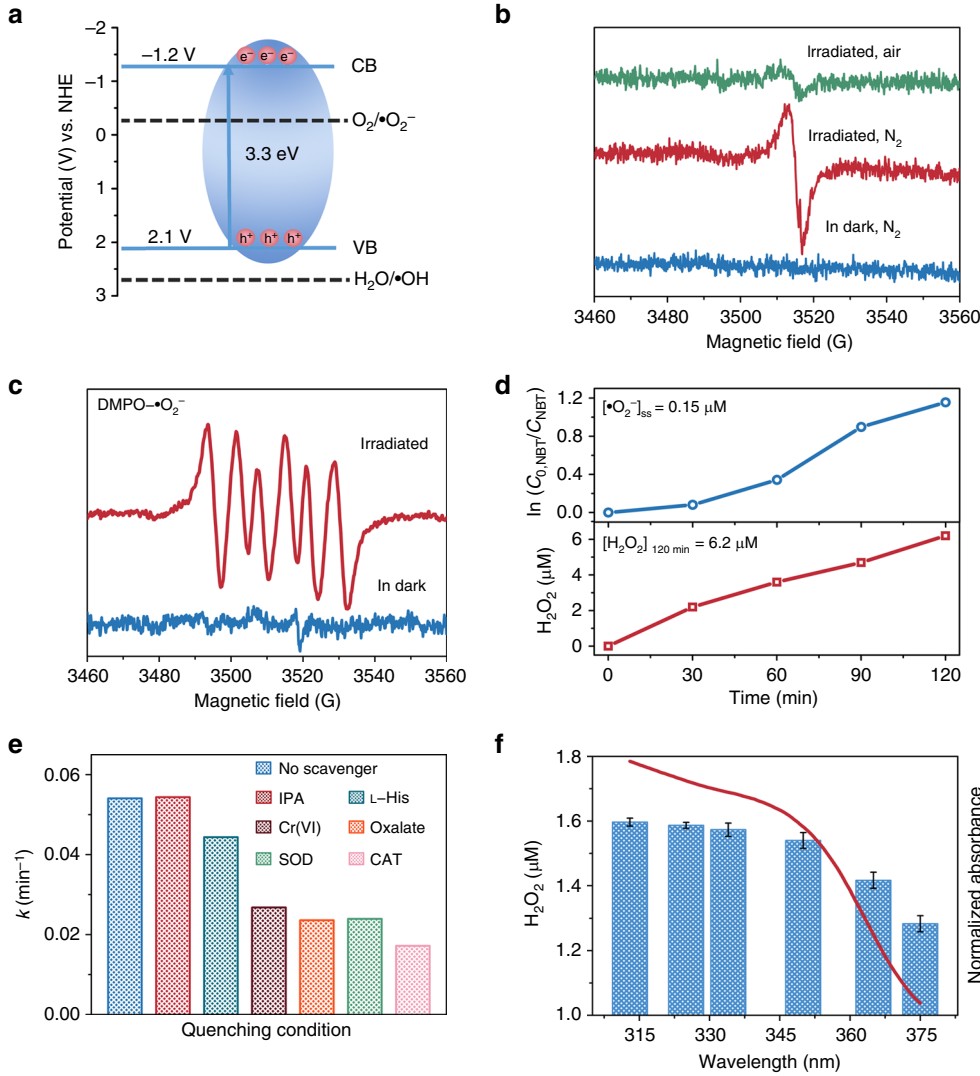

**Fig. 3** Band-structure characterization and photocatalytic disinfection mechanism of ZIF-8 (zinc-imidazolate MOF). **a** The band positions of ZIF-8 with respect to the reactive oxygen species (ROS) formation potential. Conduction band (CB) and valence band (VB) represent conduction band and valence band respectively. **b** Electron paramagnetic resonance (EPR) spectra of ZIF-8 at 77 K in dark and under light irradiation (300 nm < λ < 1100 nm) in different atmosphere. **c** EPR spectra of DMPO−•$O_2^-$ for ZIF-8 under light irradiation and in dark. **d** Steady-state concentration of •$O_2^-$ calculated from the decay of nitroblue tetrazolium (NBT) and hydrogen peroxide ($H_2O_2$) accumulation over time, respectively. **e** The first-order disinfection rate on ZIF-8 with different scavengers (IPA → •OH, L-His → $^1O_2$, Cr(VI) → e$^-$, Oxalate → h$^+$, SOD → •$O_2^-$, CAT → $H_2O_2$). **f** Dependence of the amount of released $H_2O_2$ by ZIF-8 on the wavelength of incident light and the ultraviolet−visible (UV−vis) spectra of ZIF-8. The error bars are calculated via repeating the measurements for three times. SOD superoxide dismutase, IPA isopropanol, DMPO 5,5-diemthyl-1-pyrroline N-oxide

estimated to be 3.3 eV[45]. The Mott–Schottky measurements conducted at three different frequencies give the n-type semiconductor nature of ZIF-8 and the flat band position of ZIF-8 at ~−1.2 V vs. NHE, which roughly equals to the bottom of conduction band (Supplementary Fig. 11). As shown in Fig. 3a, the valence band potential is then calculated to be 2.1 V vs. NHE, which was further proved by valence band X-ray photoelectron spectra (VB-XPS) (Supplementary Fig. 12). The band position of ZIF-8 together with ROS reaction potential was illustrated in Fig. 3a.

**Photocatalytic disinfection mechanism of ZIF-8.** Upon light excitation, photoelectrons are generated via LMCT process and trapped on the surfaces of ZIF-8 as paramagnetic $Zn^+$ sites, as supported by the obvious signal at $g = 2.003$ in EPR spectra recorded at 77 K in nitrogen (Fig. 3b)[46]. When irradiation occurs in air, EPR signal is strongly attenuated, suggesting the transfer of electrons from $Zn^+$ centers to $O_2$. Indeed, $\bullet O_2^-$ has been detected by EPR technique using spin-trap agents (5,5-diemthyl-1-pyrroline N-oxide (DMPO)), and also been quantified by employing nitroblue tetrazolium (NBT) reduction method[25,27]. As seen in Fig. 3c, six characteristic peaks for typical DMPO−$\bullet O_2^-$ adducts were obviously observed for irradiated ZIF-8, in sharp contrast to the silent spectra for unirradiated sample. As well, insoluble blue formazan was produced with the existence of NBT, and ca. 0.15 μM of steady-state $\bullet O_2^-$ was calculated correspondingly (Fig. 3d and Supplementary Fig. 13). From Fig. 3a we can see that photoelectrons in conduction band of ZIF-8 (−1.2 V vs. NHE) is negative enough to reduce the surface adsorbed $O_2$ to $\bullet O_2^-$ (−0.33 V vs. NHE)[47,48]. But the generation of $\bullet OH$ is thermodynamically unfavorable due to the deficient valence band potential of ZIF-8 (2.1 V vs. NHE) for the oxidation of water (2.68 V vs. NHE)[47], which was strongly supported by the absence of DMPO−$\bullet OH$ EPR signal for both irradiated and unirradiated ZIF-8 (Supplementary Fig. 14). The absence of $\bullet OH$ was also confirmed by using a fluorescent method with coumarin (Cou) as probe (Supplementary Fig. 15)[49]. The unchanged antibacterial performance of ZIF-8 in the presence of $\bullet OH$ quencher further indicates the incapability of ZIF-8 for $\bullet OH$ photogeneration (Fig. 3e). More importantly, by using p-hydroxyphenylacetic acid (HPA) as $H_2O_2$ probe[27], we found that $H_2O_2$ was generated in the photocatalytic system with a total concentration of ca. 6.2 μM after 120 min (Fig. 3d. and Supplementary Fig. 16). Its wavelength-dependent evolution trend matched well with the optical absorption of ZIF-8 (Fig. 3f and Supplementary Fig. 17). $H_2O_2$ was considered as the leading active species accounting for the damage of bacteria, as verified by the strongest effect of $H_2O_2$ in scavenger quenching experiments, along with the response of bacteria reduction and $H_2O_2$ level by alternating light and dark (Fig. 3e and Supplementary Figs. 18, 19). In the current system, it can be produced at the reduction site of ZIF-8 via oxygen-reduction pathway ($O_2 + 2H^+ + 2e^- \rightarrow H_2O_2$, +0.695 V vs. SHE) and the disproportionation reaction of $\bullet O_2^-$ ($2\bullet O_2^- + H^+ + H_2O \rightarrow H_2O_2 + O_2 + OH^-$, +1.44 V vs. SHE)[50]. We put forward that oxygen-reduction-related ROS ($H_2O_2$ and $\bullet O_2^-$) rather than $Zn^{2+}$ ions act as primary species contributing to photocatalytic bacterial inactivation.

**Air cleaning performance of MOFilter.** MOFilter was obtained by coating ZIF-8 nanocrystals via hot pressing of ZIF-8 precursors ($Zn(OAc)_2 \bullet 2H_2O$ and H-MeIM) and polyethylene glycol (PEG) on non-woven fabrics (NWFs) at 100 °C[51]. As evidenced by X-ray diffraction (XRD), scanning electron microscopy (SEM), and elemental mapping, ZIF-8 nanoparticles with well-defined rhombic dodecahedral shape and an average size of 80 nm were successfully generated and densely layered on the fibers of NWF (Fig. 4a, b and Supplementary Fig. 20). The MOF coatings show strong affinity to the substrate[34], as verified by bending and rubbing test (Supplementary Table 2 and Supplementary Fig. 21). And, 0.15 mg cm$^{-2}$ of ZIF-8 loading was confirmed by analyzing the weight increasement of substrates. The antimicrobial activity of MOFilter was challenged by exposing to E. coli suspension. Not surprisingly, MOFilter was endowed with contact-killing behavior, and had no measurable levels of adhered E. coli cells after 1 h of light exposure (Supplementary Fig. 22). These favorable results encouraged us to explore the applicability of MOFilter in various scenarios. First, large-scale production of MOFilter has been achieved to meet the basic need of practical applications (Supplementary Fig. 23). Then, we demonstrated the effectiveness and flexibility of MOFilter for comprehensive decontamination of air in confined environment. As illustrated in Fig. 4c and Supplementary Fig. 24, after introducing PM or bioaerosols into the inlet section divided from a chamber by MOFilter, residual PM or survived bacteria were counted by exposing collectors to the exhaust gas. As shown in Fig. 4d, MOFilter achieved 96.8% removal of $PM_{2.5}$ particles and 97.7% removal of $PM_{10}$ particles with a low pressure drop (64 Pa) at flow rate of 0.7 m s$^{-1}$, which was as good as the performance we presented detailly in our previous work[10,34]. Here we focused more on the air sterilization performance of MOFilter. Model aerosols nebulized from $10^5$ CFU mL$^{-1}$ of E. coli suspension were injected into the inlet section at flow rate of 0.3 mL min$^{-1}$ for 1 min. Noticeably, MOFilter shows highly efficient photocatalytic killing properties for E. coli dispersed in air, with bactericidal efficiency reaching to >99.99% over 30 min. That is quite distinct from the rather poor antibacterial ability of light-illuminated NWF (<89% inactivation efficiency) in control experiments. It can be seen that significant bacterial contamination occurs in the outlet section of the chamber with MOFilter in dark as gas filter (Fig. 4e). This reveals the non-toxicity of MOFilter itself against E. coli, and also reflects the negligible retention of bacteria on MOFilter. Furthermore, in nutrient broth liquid medium we incubated five pieces of MOFilter and NWF used in the above experiment, respectively (Supplementary Fig. 25). The nutrient broth medium with light-treated NWF immersed became very turbid due to the massive growth of E. coli, whereas that with light-treated MOFilter immersed was still clear with no bacteria recovery. The resulting phenomenon strongly supports that the excellent air sterilization performance of MOFilter arises from the photocatalytic bacteria-killing activity and not the physical barrier and adhesion effect. MOFilter still keeps sufficient activity after it has been continuously used for several times (Fig. 4f). MOFilter can be a promising candidate for comprehensive and deep purification of air in confined environment like home, schools, vehicles, etc.

**Antibacterial performance of MOFilter mask.** As a proof of concept, we also demonstrated the application potential of MOFilter in basic personal protective equipment like mask. MOFilter mask is a trilaminar mask with MOFilter serving as an inner biocidal layer sandwiched between two layers of NWF. The self-cleaning performance of MOFilter mask was assessed by comparison with that of commercial mask (N95). As demonstrated in Fig. 5a, the top layer of the mask was exposed to artificial pathogenic aerosols generated from E. coli suspensions for 5 min. We can see that, under 30 min of simulated sunlight illumination, the number of viable bacterial cells dropped significantly with almost no measurable levels of living bacteria survived on each layer of MOFilter mask (Fig. 5b), especially the bottom layer. This result is very favorable because the bottom layer has a direct contact with human skin. On the contrary, only

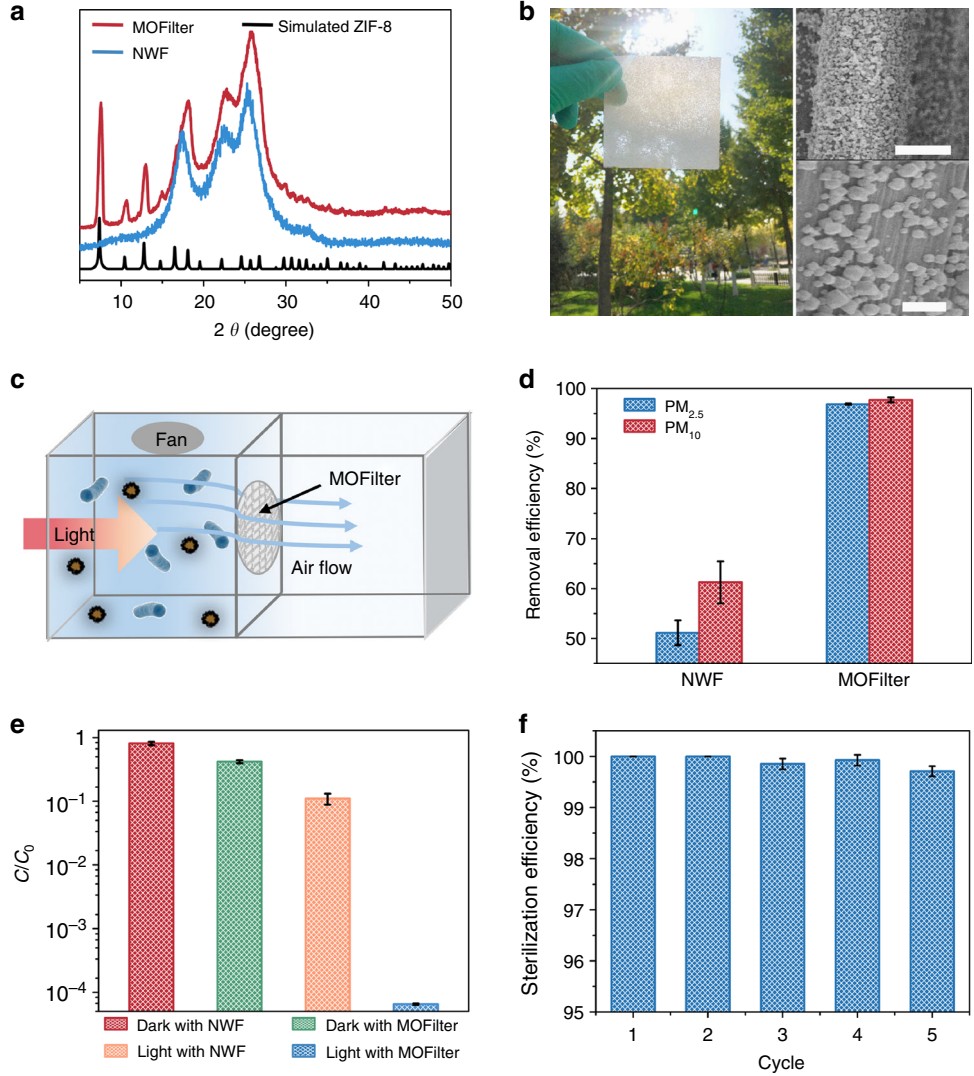

**Fig. 4** Characterization of metal-organic framework (MOF)-based filter (MOFilter) and its air cleaning performance. **a** X-ray diffraction (XRD) patterns of non-woven fabric (NWF) and MOFilter. **b** Optical photo and scanning electron microscopy (SEM) images of MOFilter (scale bar, 5 μm (top); 1 μm (bottom)). **c** Schematic representation of the air cleaning system. **d** Comparison of the particulate matter (PM) filtration efficiency between MOFilter and NWF. **e** Comparison of the air disinfection performance between MOFilter and NWF under light and dark conditions, respectively. **f** Air disinfection performance of MOFiter continuously used for five cycles. The error bars are calculated via repeating the measurements for three times

a small number of bacteria on top cover of commercial mask was killed under light in control experiment. Significant number of bacteria flourished on N95 (Fig. 5b, c). Obviously, MOFilter mask is much more effective than commercial one in pathogens prevention. Likewise, ZIF-8-based silk clothing also presents satisfactory antibacterial effect under light illumination (Supplementary Fig. 26). The remarkable bactericidal activity indicates the great protective coating application foreground of MOF.

## Discussion

In summary, we explore the photocatalytic disinfection activity of ZIF-8 and its potential use in air purification. The charge-trapping centers $Zn^+$ can be photogenerated on MOF surface via LMCT, and effectively active $O_2$ to form $\bullet O_2^-$ and related ROS like $H_2O_2$. The ROS production rather than $Zn^{2+}$ releasing mainly contributes to the biocidal properties of ZIF-8. ZIF-8 shows excellent photocatalytic antibacterial behavior, which is well beyond the performance of traditional semiconductors like ZnO and $TiO_2$. Furthermore, ZIF-8 is processed as filters with the combination of PM filtration and bacteria-killing function. We

demonstrate that ZIF-8-based filter could be a kind of strong and comprehensive protection against air hazards including PM and pathogens aerosols. This study provides valuable insights for development of porous photocatalytic antibacterial materials and also open the door for the application of these materials in air disinfection.

## Methods

**Material characterization**. Powder X-ray diffraction (XRD) patterns of the samples were measured by a Bruker Focus D8 diffractometer operating at 40 kV voltage and 50 mA current with Cu-Kα X-ray radiation ($\lambda = 0.154056$ nm). Concentrations of $Zn^{2+}$ were determined by ICP-MS with Agilent 7700 spectrometer. Low-temperature EPR measurements were performed on a BrukerA300 spectrometer operating at an X-band frequency of 9.85 GHz. All spectra were acquired at 77 K. EPR signals of spin-trapped paramagnetic species with DMPO were recorded with a Bruker E500 spectrometer. UV–vis spectra were recorded on a UV-2600 spectrometer (SHIMADZU). The Fourier-transform infrared spectroscopy (FT-IR) spectra were recorded in the range 400–4000 $cm^{-1}$ on Nicolet 170 SXFT/IR spectrometer. The thermogravimetric analysis (TG) was carried out using a TA Instruments STA449F5 apparatus in the temperature range of 35–800 °C under $N_2$ flow at a heating rate of 10 °C $min^{-1}$. $N_2$ sorption tests were measured using a Builder SSA-4200 automatic volumetric gas adsorption analyzer. Field-emission SEM was performed on a JEOL model JSM-7500 F with an accelerating voltage of

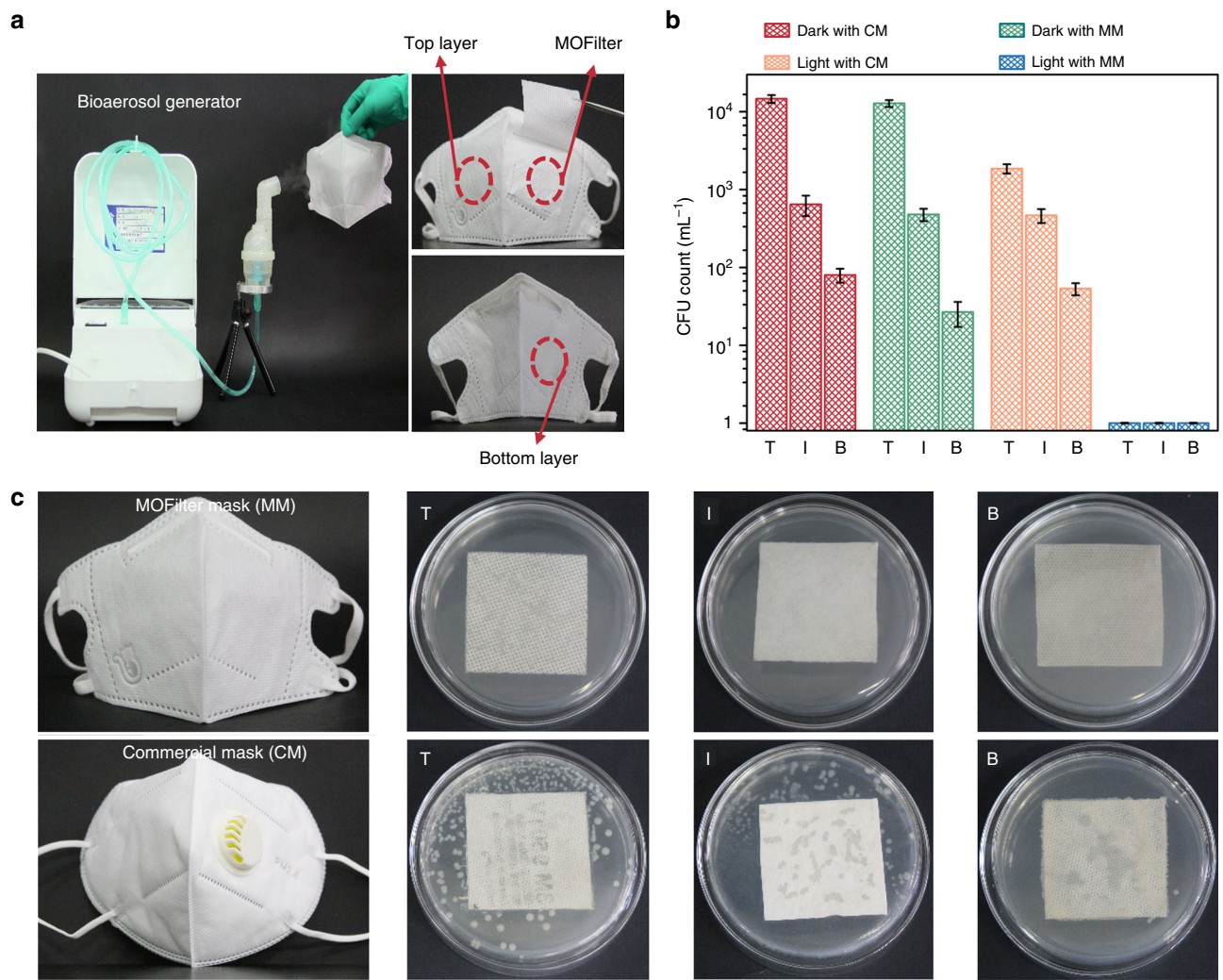

**Fig. 5** Antibacterial performance comparison between metal-organic framework (MOF)-based filter (MOFilter) mask (MM) and commercial mask (CM). **a** Bioaerosol generation apparatus and optical images of trilaminar MOFilter mask. **b**, **c** *Escherichia coli* levels residual on top, inner and bottom layers of MM and CM, respectively, after 30 min of light irradiation. The number of viable cells in **b** is determined from saline eluent used for collecting living bacteria on each layer of mask after reaction. The bacterial colonies residual on eluent-treated mask are shown in **c**. The top, inner, and bottom layers are denoted as T, I and B, respectively. The error bars are calculated via repeating the measurements for three times

5.0 kV and a current of 10 mA. Transmission electron microscopy (TEM) was carried out by a JEOL JEM-1200EX microscope. Fluorescence images of bacterial cells stained with N01 dye and PI dye were captured by Olympus FV1000 confocal fluorescence microscope. VB-XPS was performed on a Thermo Fisher ESCALAB 250Xi using a monochromatic Al Kα X-ray source. Mott–Schottky measurements were performed in 0.5 M $Na_2SO_4$ solution by using a CHI 760E electrochemical workstation (Shanghai, China) with a standard three-electrode cell. The working electrodes were prepared as follows: the as-prepared photocatalyst (7 mg) was ultrasonically dispersed in 0.5% Nafion solution (0.5 mL) to produce a slurry and then dip-coated onto a FTO glass electrode ($2 \times 0.5$ cm$^2$). Pt plate and Ag/AgCl electrode were used as counter electrode and reference electrode, respectively.

**Synthesis of MIL-100(Fe).** One point one hundred and ninety nine grams of Fe(NO$_3$)$_3$·9H$_2$O, 0.4129 g of trimethyl 1,3,5-benzenetricarboxylate (BTC), 0.2976 mL of hydrogen fluoride solution (40 wt%), and 15 mL of $H_2O$ were put into a 20 mL Teflon-lined autoclave and heated at 150 °C for 84 h. After cooling the mixture to room temperature, the orange crystals were centrifuged. A treatment in hot deionized water (80 °C) for 3 h and followed by hot ethanol (60 °C) for 1 h was applied to decrease the amount of residual BTC. The final products were washed with ethanol and dried vacuum at 80 °C for 24 h[52].

**Synthesis of NH$_2$-MIL-125(Ti).** 2-Amino-1, 4-benzenedicarboxylic acid (NH$_2$-BDC) and *p*-toluylic acid (p-TA) with a molar ratio of 1:4 were chosen as organic linkers, and dissolved into a dimethylformamide (DMF)-methanol mixed solvent (v(DMF)/v(methanol) = 9:1). Next, titanium isopropoxide (TPOP) was added into the above solution and stirred for several minutes, the molar ratios of TPOP and

NH$_2$-BDC is 2:3.1. Then, the reactant mixtures were moved to a 100 mL Teflon-lined steel autoclave, sealed, and placed in an oven at 150 °C for 24 h under static conditions. After that, the product was washed for several times with DMF and methanol separately to remove the remaining reactants after cooling. Then, the samples were dried overnight in a vacuum oven at 60 °C[53].

**Synthesis of NH$_2$-UiO-66.** ZrCl$_4$ (pre-dissolved in a DMF/HCl mixture of v (DMF)/v(HCl) = 5:1) and 2-amino-1, 4-benzenedicarboxylic acid (pre-dissolved in DMF) at a molar ratio of 1:1.4 were mixed and heated at 80 °C overnight. The obtained powders were isolated by centrifugation and washed with DMF (3 × 30 mL). Then, the powders were immersed in ethanol for 3 days (the solvent was replaced with fresh ethanol each day). Finally, the product was dried under vacuum at 150 °C overnight[54].

**Synthesis of ZIF-11.** A solid mixture of Zn(NO$_3$)$_2$·4H$_2$O (0.60 g) and benzimidazole (H-PhIM) (4.2 g) was dissolved in 360 mL diethylformamide (DEF) in a 500 mL wide-mouth glass jar. The capped jar was heated for 4 days in an isothermal oven at 100 °C. The jar was then removed from the oven, and allowed to cool to room temperature naturally. The powder and mother liquor were removed by repeating the cycle of decanting, washing with DMF, sonicating several times, and dried in the vacuum oven at 120 °C for 12 h[55].

**Synthesis of ZIF-8.** One point two hundred and ninety one grams of Zn(NO$_3$)$_2$·4H$_2$O was dissolved in 100 mL of methanol, and 1.621 g of 2-H-MeIM was dissolved in 100 mL of methanol, respectively. Then, the two solutions were mixed

under stirring and kept statically at room temperature for 24 h. The resultant powders were collected by centrifugation and thoroughly washed with methanol. The product was dried at 120 °C for 12 h in a vacuum oven[56]. Highly crystallized and phase pure ZIF-8 particles have been prepared as confirmed by PXRD results (Supplementary Fig. 1). ZIF-8 particles were also characterized by FT-IR, TG, TEM, SEM, elemental mapping, and Brunauer–Emmett–Teller (BET) adsorption (Supplementary Fig. 2). The as-obtained ZIF-8 nanocrystals show typical rhombic dodecahedron morphology, and have a high BET specific surface area of 1538 m$^2$ g$^{-1}$. The particle size of ZIF-8 ranges from 70 to 110 nm, and an average diameter of 92 nm was obtained by statistical evaluation of 100 objects.

**Fabrication of MOFilter.** Four grams of zinc acetate dihydrate (0.02 mol), 10 g of 2-H-MeIM (0.12 mol), and 2 mL of PEG (Mn = 200) were manually ground and mixed. The mixture was then loaded on substrate (the substrate is NWFs, unless otherwise specified), packed with aluminum foil, and heated with electric heating plate at 100 °C for 2 min. After peeling off the aluminum foil, the slice was fully washed with ethanol, dried at 80 °C for 30 min and activated at 120 °C for 6 h prior to use as MOFilter[34].

**Photocatalytic disinfection activity of MOFs in solution.** Gram-negative E. coli was used as model bacteria. All disks and materials were sterilized in an autoclave before experiments. The bacterial cells were grown in nutrient broth at 37 °C for 18 h to yield a cell count of approximately 10$^9$ CFU mL$^{-1}$. Then, bacterial cells were collected by centrifugation (5000 rpm for 10 min) and resuspended in sterile saline solution (0.9% (w/v)). The bacteria concentration for bactericidal study was 10$^7$ CFU mL$^{-1}$, which was adjusted by gradient dilution method using 0.9% (w/v) saline solution.

Typically, 5 mg of catalyst was added to a 50 mL photoreactor containing 10 mL of bacteria solution (10$^7$ CFU mL$^{-1}$). The bacteria and photocatalyst were mixed by using a magnetic stirrer at room temperature, and simultaneously irradiated by simulated sunlight (300 W Xe lamp coupled with an AM 1.5 filter) for 120 min at the density of 100 mW cm$^{-2}$. As the reaction proceeded, the mixture was carefully pipetted out at a scheduled interval and the residual bacteria concentrations were determined by the standard plate count method. The plates were incubated at 37 °C for 20 h. The number of colonies was enumerated through visual inspection. There were a series of experiments conducted in the dark at the same situations as the dark controls. The light control group was carried out in the absence of photocatalyst.

**Regrowth test.** After photocatalytic disinfection experiment, the reaction solution of ZIF-8 was stirred in dark at slow rate of 200 rpm at 10 min, 24 h, and 48 h, respectively. Then, 400 μL of the solution at different time intervals was added to 3.6 mL of nutrient broth liquid medium and the mixture was incubated at 37 °C for 20 h in a shaker. The optical density at 600 nm was monitored every 6 h.

**Antibacterial activity of Zn$^{2+}$.** The MIC and MBC methods were applied to assess the antibacterial activity of Zn$^{2+}$. In MIC test, the serial Zn$^{2+}$ solutions with different concentrations were dispersed in sterilized tubes with 2 mL broth media by a two-fold serial dilution method. Then, 20 μL bacteria suspensions (10$^7$ CFU mL$^{-1}$) were added into the serial tubes, respectively. Finally, the tubes were incubated at 37 °C for 20 h and the lowest concentration of samples that inhibited visible growth of bacteria by turbidimetric method were signed as the MIC value.

MBC is defined as the minimum concentration of the sample required to kill 99.9% bacteria after a defined period of incubation. In MBC test, the nutrient agar was spread onto a Petri plate, and then the invisible bacterial suspensions with different concentrations of samples taken out from the tubes were coated on the agar plates. Final agar plates with bacterial suspension were incubated at 37 °C for 20 h. The number of survival colonies was counted to get the MBC of Zn$^{2+}$.

**Photocatalytic disinfection activity of MOFilter.** The photocatalytic measurements were conducted according to Chinese standard GB/T 30706-2014 procedure with a slight modification. Two milliliters of bacteria suspension (10$^7$ CFU mL$^{-1}$) was dropped on MOFilter surface gradually, and then irradiated by simulated sunlight for 1 h at the density of 100 mW cm$^{-2}$. After photocatalytic disinfection reaction, MOFilter was fully washed with 20 mL of 0.9% (w/v) saline solution and the resultant eluant was dispersed in sterilized tubes with 2 mL saline solution by a two-fold serial dilution method. Then, the samples (50 μL) were plated on nutrient agar culture medium. These plates were incubated at 37 °C for 20 h. Then, the viable cell count was performed to obtain the results for disinfection. Also, MOFilter freshly washed by eluant was cultured in nutrient agar for 20 h at 37 °C for residual analysis of adhered viable cells. There were a series of experiments conducted in the dark at the same situations described above. The light control group was carried out in the absence of photocatalyst. All the experiments were repeated for three times.

**Photocatalytic air disinfection.** A rectangular chamber with a cross-section of 30 × 30 cm$^2$ (height × width) and a length of 60 cm was assumed to be air duct model, which is divided into an inlet section, a reaction section, and an outlet

section by MOFilter with 8 cm long and 8 cm wide. The whole device was located in a class II microbiological safety cabinet to ensure sterility of the experiments. Escherichia coli-containing aerosols with particle diameter from 1 to 5 μm, which was of a similar size to aerosols generated by a human sneezing or coughing, were prepared as model aerosols by nebulizing 10$^5$ CFU mL$^{-1}$ of E. coli suspension was sprayed into the reaction section at the flow rate of 0.3 mL min$^{-1}$ for 1 min. MOFilter was exposed to them under simulate sunlight illumination for 30 min and then air in the reaction section was sampled at a flow rate of 28.3 L min$^{-1}$ with an air impactor through six 400-hole sieved head plates and over 90 mm Petri dishes. The sampling volume was adjusted to 28.3 L to avoid plate saturation. The petri dishes attached to the sampler heads contained growth medium for bacteria. Bacterial samples were collected using nutrient agar and incubated at 37 °C for 20 h. Every test was repeated three times. After incubation, the colonies were counted. The photocatalytic air disinfection activity of NWFs was evaluated under the same conditions described above by replacing MOFilter with NWF. The concentration of microorganisms in the air, expressed in CFU m$^{-3}$, was calculated using the following Eq. (1):

$$[\text{bioaerosol}] = \frac{N}{Ft},$$

where "F" and "t" are the flow rate and the sampling time, respectively, and "N" is the colony counts after positive hole correction.

The air disinfection efficiency of MOFilter was calculated according to the following Eq. (2):

$$\text{Efficiency} = \frac{(C_0 - C)}{C_0},$$

where "$C_0$" (CFU m$^{-3}$) represents the bacterial concentration dispersed in air of reaction section after introducing E. coli–containing aerosols immediately.

**PM removal experiment.** PM removal test followed the method previously reported by our group. A piece of filter with a diameter of 4 cm was set at one side of the pipe, and an electric fan was put on the other side to help the air pass through the filter at a constant velocity (0.7 m s$^{-1}$) measured by an anemograph (EDKORS, FS-801). The filtered air was collected in a plastic bag, and a particle counter was used to detect the PM mass concentrations with and without the filter. The detection was finished before the plastic bag reached its maximum volume, and this constant pressure process ensured that the concentration of PM remained unchanged during collection. Twenty groups of mass concentration data were collected to give the average concentration. All the PM removal tests were conducted in triplicate. The PM removal efficiency of the MOFilter was calculated with the following Eq. (3):

$$\text{Efficiency} = \frac{(C_0 - C)}{C_0},$$

where "$C_0$" (μg m$^{-3}$) and "$C$" (μg m$^{-3}$) are the mass concentrations of particle matter tested with and without the filter.

**Photocatalytic antibacterial activity of MOFilter mask.** Escherichia coli-containing aerosols with the diameter of 1–5 μm generated from 10$^6$ CFU mL$^{-1}$ of bacteria suspension were prepared as model-infected aerosols. The top layer of MOFilter mask was exposed to a 0.3 mL min$^{-1}$ of aerosols flow for 5 min and then irradiated by simulated sunlight for 30 min at the density of 100 mW cm$^{-2}$. After that, each layer of MOFilter mask was fully washed with 20 mL 0.9% (w/v) saline solution, respectively. The concentrations of bacteria in eluant were determined by the standard plate count method. Also, each layer of mask freshly washed by eluant was incubated in nutrient agar for 20 h at 37 °C for residual analysis of adhered viable cells. There were a series of experiments conducted in the dark at the same situations described above. The antibacterial performance of commercial mask was assessed under the same conditions described above. All the experiments were repeated for three times.

**Empirical disinfection kinetics.** The kinetics of photocatalytic bacterial inactivation are usually described using empirical equations. The Chick–Watson equation is the classical model for microorganism inactivation with a constant concentration of disinfecting agent, as expressed by following Eq. (4):

$$\lg\left(\frac{C}{C_0}\right) = -k[\chi]^n t,$$

where "$C_0$" and "$C$" are the initial and subsequent bacterial concentrations, respectively, at the beginning of the process and after time 't' respectively, "$k$" is the disinfection kinetic constant, "$\chi$" is the concentration of disinfecting agent (photocatalyst in this case), and "$n$" is the order of reaction.

**Fluorescent-based cell live/dead test.** The cell membrane damage of the bacterial cells was determined with fluorescence microscope. One milliliter of bacteria and ZIF-8 slurry during photocatalytic treatment was collected, centrifuged, and washed with 0.9% (w/v) saline solution, followed by staining with dyes of LIVE/DEAD BBcellProbe$^{\text{TM}}$ N01/PI (BestBio Inc., China) bacterial viability kit according to the manufacturer's protocol. The experiment was carried out using a mixture of

N01 dye and propidium iodide (PI) dye. Bacterial cells with intact cell membrane (live) are stained by N01 and fluorescent green, whereas PI penetrates only damaged membranes and stains the dead bacterial cells. After incubation in dark for 15 min, the stained samples were observed by a fluorescence microscope (Olympus, FV1000) with ×10 magnification.

**ROS measurements**. $\bullet O_2^-$ steady-state concentration was calculated by measuring the decay of NBT (Sigma, 98%) using UV–vis spectroscopy. NBT has an absorption peak at 260 nm. The rate constant for $\bullet O_2^-$ and NBT reaction is $5.9 \times 10^4 \, M^{-1} \, s^{-1}$ [25,57]. $H_2O_2$ concentration was measured using a HPA fluorescence probe. A fluorescent p-hydroxyphenylacetic acid dimer is formed by the reaction of $H_2O_2$ with HPA using horseradish peroxidase as a catalyst. The amount of the dimer is analyzed using a fluorescence spectrophotometer at the emission wavelength of 410 nm with the excitation at 310 nm. $\bullet OH$ was determined by the fluorescence method using Cou as a probe molecule. One millimolar Cou and 0.5 mg $mL^{-1}$ ZIF-8 were dispersed in 20 mL aqueous solution. Fluorescence spectra of generated 7-hydroxycoumarin, which could emit fluorescence at 455 nm when excited at 332 nm, was measured. Samples were collected at different time intervals and filtrated by 0.22 μm membrane to exclude the influence of bacteria cells and photocatalysts.

**Scavenger quenching experiments**. The scavengers used was sodium chromate (Cr(VI)), 2.5 mM, Sigma, 99.5%) for electron, superoxide dismutase (400 U $mL^{-1}$, Sigma, 99%)) for $\bullet O_2^-$, L-histidine for $^1O_2$ (L-His, 2.5 mM, Sigma, 99%), catalase for $H_2O_2$ (300 U $mL^{-1}$, Sigma), isopropanol (2.5 mM, Sigma, 99.5%) for $\bullet OH$ and sodium oxalate (2.5 mM, Sigma, 99.5%) for hole. The scavengers were added into the bacteria suspension before illumination. The concentrations of bacteria in solution were measured at different time intervals by using standard spread plating techniques. Each sample was serially diluted and each dilution was plated in triplicate onto nutrient agar and incubated at 37 °C for 20 h.

## Data availability

The data sets generated during and/or analyzed during the current study are available from the corresponding authors on reasonable request. The data behind Figs. 2–5 are available in the Supplementary source data file.

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

## Acknowledgements

This work was financially supported by the National Natural Science Foundation of China (Grant Nos. 2165102, 21471018, 21490570, 21674012, 21801017), Beijing Municipal Science and Technology Project (Z181100004418001), Beijing Natural Science Foundation (2184121), and Beijing Institute of Technology Research Fund Program.

## Author contributions

P.L. synthesized the catalysts and conducted all the structure analysis and the photo-catalytic studies. J.Z.L., X.F., J.L., Y.C.H., J.W.Z., H.W., A.X.Y. and J.W.Z. assisted with the material characterizations and catalysis measurements. B.W. and X.J.M. co-wrote the paper. B.W. and X.J.M. supervised the research. All authors discussed the results and assisted during manuscript preparation.

## Additional information

**Competing interests:** The authors declare no competing interests.

