## [Peer Review File · Nature Communications]

Reviewers' comments:

Reviewer #1 (Remarks to the Author):

This paper presents a work on preparation of ZIF-8 and its application on a nonwoven fabric as air filter material with photoactive antibacterial functions. The intended application of the materials is interesting, but the paper is lack of some fundamental information for proposed applications.

UV sterilization is an efficient method in providing antibacterial functions. On page 2 the authors stated UV...has 'ow antibacterial efficiency". Please provide evidence.

What are the size information and characterizations of ZIF-8 particles? When zinc acetate dihydrate and 2-methylimidazole were mixed in PEG and applied onto fabrics, any characterization of the produced particles in the solution and on surfaces of the fibers?

No details on nonwoven fabric used in this work. Which polymer, fiber size, made by which process? Density, porosity...?

SEM images of particles on surface of fibers reveal some surface adhesion but no penetration into the fibers, which bring in a concern on falling-off of the particles from the surfaces, especially under frictions. Please provide some durability testing results.

ZIF-8 needs 120 min to provide 99.9999% reduction of E. coli in solution containing 5mg/10mL, pretty high concentration.

The produced H₂O₂ could still stay in the solution and provide antibacterial effect. How do you differentiate the kill provided during and after the UV exposure.

The UV lamp of 300W Xe shines a light in intensity similar to direct sunlight. It could already kill a significant amount of E. coli.

In Fig 2d, what are the sizes of TiO₂ and ZnO particles? Here, what is the size of ZIF-8 again?

How to assign the broad absorption at 350nm of ZIF-8 as LMCT? I couldn't find any related information from the reference #46.

Based on XRD, ZIF-8 is a crystal, but Tauc plot is often used for amorphous materials not for crystal.

Is Zn²⁺ or ZIF-8 serve as photosensitizer?

If H₂O₂ is produced in the system, antibacterial tests should use a quenching chemical to remove it during tests.

Reviewer #2 (Remarks to the Author):

Review for manuscript Manuscript NCOMMS-18-37114-T:

Title : "Metal-organic frameworks with superior photocatalytic bactericidal activity for integrated air cleaning"

Authors:

Ping Li, Jiazhen Li, Xiao Feng, Jie Li, Yuchen Hao, Jinwei Zhang, Hang Wang, Anxiang Yin, Junwen Zhou, Xiaojie Ma*, Bo Wang*

The manuscript by Ping Li et al. describes the fabrication of antibacterial and self-cleaning filtration materials. The photocatalytic bactericidal activity of five typical water stable MOFs with significant emphasis on ZIF-8, that outperforms the extensively used biocidal photocatalysts ZnO and anatase TiO₂. The authors found out that oxygen-reduction related reactive oxygen species (H₂O₂ and •O₂⁻) rather than Zn²⁺ ions act as primary source contributing to photocatalytic bacterial inactivation. The results are well supported by EPR studies, via scavenger quenching experiments, as well as employing nitroblue tetrazolium reduction method along with Zn²⁺ leaching tests and numerous control experiments. The authors demonstrated large-scale production of MOFilter and their use in basic personal protective equipment like mask.

Comments:

Overall, the manuscript was very interesting to read and the authors made a nice efforts to understand and characterize the underlying mechanism for photo catalytic properties of ZIF-8. However, there are different examples in the literature on the photocatalytic and antibacterial properties of MOF and particularly ZIF-8, which present different conclusions. In my opinion these papers should be mentioned and the different outcome discussed:

1. The authors claim as, "For the first time, the photocatalytic activity of ZIF-8 was discovered and characterized", I would suggest to rephrase this statement.

Photocatalytic activity of ZIF-8 is described in different setup in the literature such as, "The active roles of ZIF-8 on the enhanced visible photocatalytic activity of Ag/AgCl: Generation of superoxide radical and adsorption" (Journal of Alloys and Compounds 693 (2017) 543-549); "Photocatalytic degradation of methylene blue in ZIF-8 (RSC Adv., 2014, 4, 54454); "Rapid Construction of ZnO@ZIF-8 Heterostructures with Size-Selective Photocatalysis Properties (ACS Appl. Mater. Interfaces 2016, 8, 9080–9087), Synthesis of Au@ZIF-8 single- or multi-core–shell structures for photocatalysis, Chem. Commun., 2014, 50, 8651–8654).

2. Biocidal activity of ZIF-8 is recently reported "Self-Cleaning and Antibacterial Zeolitic Imidazolate Framework Coatings"(Adv. Mater. Interfaces 2018, 5, 1800167), where the authors conclude that the antibacterial activity stems from 2-methylimidazole (H-MeIM) and zinc (Zn²⁺) ions that were released from ZIF-8 particles. On contrary, in current study, Zn²⁺ and H-MeIM had little antibacterial effect (Fig. 2c). It would be interesting if the authors could explanation their contradictory observation with reference to the given example or any other if they find in the literature. Alternatively the authors should clearly state the different outcome and which experiments confirm their hypothesis.

3. For ZIF-8, the estimated bang gap of 3.3 eV (this study) is different from previously published value of 4.9 eV (Angew. Chem. Int. ed. 2011, 50, 450). It would be helpful if the authors explain their different finding compared to the literature values. A clear input would help to avoid discrepancies in the literature or misunderstanding of the values provided.

4. The core message of this study reports reactive oxygen species mainly contribute to the antibacterial properties rather than the Zn ions, supported and evidenced principally by EPR studies. With respect to the estimated band gap, statement of generation of hydroxyl radical is thermodynamically unfavorable (as also supported via scavenger experiments). This is confusing as another report (RSC Adv., 2014, 4, 54454) describes ZIF-8 photocatalysts exhibiting efficient photocatalytic activity for methylene blue degradation under UV irradiation, which was confirmed through the detection of hydroxyl radicals (.OH) by a fluorescence method.

In my opinion the article is very interesting to the readers of Nature Communications and the study is and conclusions are well supported by experiments. However, in order to understand and correlate the new findings, appropriate and directly related citations are missing (few examples are discussed above), which should be added and discussed.

Responses to referees' comments

Referee 1:

1. *With regard to the comment “UV sterilization is an efficient method in providing antibacterial functions. On page 2 the authors stated UV...has ‘low antibacterial efficiency’”. Please provide evidence.”*

UV sterilization is indeed an efficient method, especially UVC (200-280 nm) germicidal irradiation from artificial light source. However, in ground-level solar spectrum, UV irradiation (mostly UVA and UVB) only accounts for 4% of the total energy. Under sunlight, sterilization is usually inefficient and time-consuming. In this work we introduce a highly efficient solar-assisted catalytic disinfection as a viable alternative to other disinfection methods.

Now we have clarified the above points in the revised manuscript (Page 2, Line 26-28).

2. *Concerning the comment “What are the size information and characterizations of ZIF-8 particles? When zinc acetate dihydrate and 2-methylimidazole were mixed in PEG and applied onto fabrics, any characterization of the produced particles in the solution and on surfaces of the fibers?”*

In our original paper, ZIF-8 particles have been characterized by PXRD, UV-vis diffuse reflectance spectra and EPR (Fig. S6, S10 and 3b). The PXRD results confirmed that highly crystallized and phase pure ZIF-8 particles have been prepared. Per request, we now have thoroughly characterized all samples using FT-IR, TG, TEM, SEM, elemental mapping and BET adsorption in the revised

supplementary information (Fig. S2). SEM and TEM images reveal that the as-obtained ZIF-8 nanocrystals show typical rhombic dodecahedron morphology. The particle size of ZIF-8 ranges from 70 to 110 nm, and an average diameter of 92 nm was obtained by statistical evaluation of 100 objects.

ZIF-8 coating layers were fabricated in-situ on the non-woven fabrics. They have been characterized by using PXRD (Fig. 4a), SEM and elemental mapping (Fig. S20). The PXRD pattern of MOFilter confirmed the successful formation of a dense layer of ZIF-8 on the substrates, which was further evidenced by SEM and elemental mapping. ZIF-8 nanocrystals grown on the fibers also show well-defined rhombic dodecahedral shape. An average size of 80 nm for ZIF-8 particles on fibers was obtained by statistical evaluation of 100 particles.

Accordingly, we have included all the above results and related discussions in the revised manuscript (Page 8, Line 15-17) and the supplementary information (Page 4, Line 1-10; Page5-6, Line 27-30, 1-2 and Fig. S2 and S20).

3. *As to the question “No details on nonwoven fabric used in this work. Which polymer, fiber size, made by which process? Density, porosity...?”*

Non-woven fabric employed in this work is the spunlaced non-woven fabric with polyethylene (PE) and polyethylene terephthalate (PET) (1:2). It has a fiber diameter of ca. 12 μm , thickness of 0.190 mm, density of 1.4 g cm^{-3} , maximum pore size of 116.6 μm , average pore size of 56.1 μm and porosity of 69.7%.

Per request, we have included the above detailed information in the revised supplementary information (Page 3, Line 8-12).

4. *As for the comments “... a concern on falling-off of the particles from the surfaces, especially under frictions. Please provide some durability testing results.”*

Per request, we have studied the durability of MOFilter *via* bending and rubbing test. Specifically, 10 pieces of MOFilter were continuously bended and rubbed for 1000 times, and after that the changes of samples were carefully measured and recorded. As shown in Table S2, there were no significant changes in MOFilter weight, indicating the strong affinity of ZIF-8 nanocrystals to the substrate. Moreover, no obvious changes were observed for surface morphology of fibers after the above-mentioned tests (Fig. S21). The high robustness of ZIF-8 coatings had also been verified by abrasion

resistance tests and mechanical stirring experiment in our previous work (*Angew. Chem. Int. Ed.*, 2016, 55, 3419–3423).

To further clarify this point, we have now added the above results and discussions in the revised manuscript (Page 8, Line 18-19) and the supplementary information (Table S2 and Fig. S21).

5. *Regarding the suggestion “ZIF-8 needs 120 min to provide 99.9999% reduction of E. coli in solution containing 5mg/10mL, pretty high concentration.”*

Per suggestion, we also achieved 99.999% reduction of *E. coli* in solution by lowering the catalyst dosage to 300 mg L⁻¹ (Fig. S3). More importantly, here we do need to use sufficient amount of ZIF-8 to achieve high PM aerosol capture efficiency while maintaining the excellent surface bactericidal performance (Fig. 4d, 4e and 4f). Comparing with other filters with inorganic sorbents (eg. PP/PET non-woven with activated carbon), MOFilter has a drastically lower MOF loading (0.15 mg cm⁻²). It is noteworthy that the volumetric concentration of the photocatalyst in aqueous solution is more relevant in water purification. In this work, we mainly focus on the application of these bifunctional MOFilters with impressive self-cleaning and anti-biofouling effects for integrated air cleaning.

We have now included the above results and discussions in the revised manuscript (Page 5, Line 7-9) and the supplementary information (Fig. S3).

6. *As to the inquiry “The produced H₂O₂ could still stay in the solution and provide antibacterial effect. How do you differentiate the kill provided during and after the UV exposure.”*

To differentiate the bacteria inactivation under light and dark conditions, we have now investigated the response of *E. coli* viability as well as the concentration change of formed H₂O₂ by alternating light and dark every 30 min in the presence of ZIF-8. As shown in Fig. S19, light-initiated H₂O₂ generation resulted in a dramatic reduction of bacteria due to the ROS disinfection mechanism proposed in our manuscript. Most of the generated H₂O₂ has been consumed by bacteria in this process. During the dark period bacteria reduction was inhibited obviously since no more H₂O₂ was produced in such condition.

Fig. S19. Disinfection performance of ZIF-8 and the concentration of generated H_2O_2 versus time under the alternated light and dark conditions (irradiation in white and dark periods in gray).

We have now included the above results in the revised manuscript (Page 7, Line 24-25) and the supplementary information (Fig. S19).

7. *As for the comments “The UV lamp of 300W Xe shines a light in intensity similar to direct sunlight. It could already kill a significant amount of E. coli.”*

In this work, a 300 W Xe lamp coupled with a AM 1.5 filter ($300\text{ nm} < \lambda < 1100\text{ nm}$) was employed as light source with optical power density fixed at 100 mW cm^{-2} (1 sun) to simulate sunlight irradiation. Under this light illumination, a certain amount of *E. coli* can indeed be killed without photocatalyst as seen in Fig. 2b. However, only less than 77% inactivation efficiency ($0.6, -\log_{10}(C/C_0)$ efficiency) was achieved within 120 min. In contrast, $> 99.9999\%$ efficiency ($6.1, -\log_{10}(C/C_0)$ efficiency) was obtained with the introduction of ZIF-8 under identical sunlight irradiation.

To further clarify this point, we have presented a detailed discussion in the revised manuscript (Page 5, Line 1-2).

8. *In response to the question “In Fig 2d, what are the sizes of TiO_2 and ZnO particles? Here, what is the size of ZIF-8 again?”*

Per request, we have included the SEM and TEM images of TiO_2 , ZnO and ZIF-8 along with the size distribution of these particles in the revised supplementary information (Fig. S8). A statistical evaluation shows that TiO_2 , ZnO and ZIF-8 particles are about 15-35 nm, 20-100 nm and 70-110 nm in diameter respectively, with an average size of about 25 nm, 50 nm and 92 nm, respectively.

9. Regarding the suggestion “How to assign the broad absorption at 350nm of ZIF-8 as LMCT? I couldn't find any related information from the reference #46.”

ZIF-8 shows strong absorption at 227 nm and a broad absorption at longer wavelength of 350 nm (Fig. S10). The strong absorption centered at 227 nm could be attributed to intraligand charge transfer, and shows a slight red shift relative to 2-methylimidazole (H-MeIM). After illuminating ZIF-8 with $300\text{ nm} < \lambda < 1100\text{ nm}$ light, an obvious signal emerged in EPR spectra (Fig. 3b) due to the generation of trapped electrons on Zn^+ sites *via* LMCT process. This indicates that the broad absorption at 350 nm is responsible for such electron transfer to the metal center. As previously reported for MOF-5 (*J. Phys. Chem. B.*, 2006, 110, 13759-13768; *Chem. Commun.*, 2004, 20, 2300–2301), the presence of structural Zn_4O_{13} clusters was responsible for ligand to metal charge transfer centered at 350 nm.

Fig. S10. UV-vis diffuse reflectance spectra of ZIF-8 and H-MeIm.

We have included the above discussions and related figures in the revised manuscript (Page 6, Line 26-31) and the supplementary information (Fig. S10).

10. Concerning the comment “Based on XRD, ZIF-8 is a crystal, but Tauc plot is often used for amorphous materials not for crystal.”

Indeed, Tauc plot has been used for the evaluation of the band gap of some crystalline materials (*J. Am. Chem. Soc.*, 2016, 138, 13822–13825; *Nature Commun.*, 2018, 9, 1660-1668). Per suggestion, we have also calculated the optical band gap of ZIF-8 by using simple equation (band gap = $1240 / \text{wavelength}$). A value of 3.3 eV was obtained, which is the same as the result derived from Tauc plot.

11. As for the comments “Is Zn^{2+} or ZIF-8 serve as photosensitizer?”

ZIF-8 with optical absorption centered at 350 nm due to LMCT serves as the photosensitizer. Zn^{2+} with d^{10} electronic configurations has no obvious absorption in the UV and visible region, and could not harvest light.

12. *As to the inquiry “If H_2O_2 is produced in the system, antibacterial tests should use a quenching chemical to remove it during tests.”*

We did conduct quenching experiment to determine the bactericidal contribution for each reactive oxygen species (ROS) (Fig. 3e and Fig. S18). The scavengers used was sodium chromate (Cr(VI)) for electron, superoxide dismutase (SOD) for $\bullet O_2^-$, L-histidine for 1O_2 (L-His), catalase for H_2O_2 (CAT), isopropanol (IPA) for $\bullet OH$ and sodium oxalate for hole. The scavengers were added into the bacteria suspension before illumination. According to the experiment results, we propose that oxygen-reduction related ROS (H_2O_2 and $\bullet O_2^-$) act as primary species contributing to photocatalytic bacterial inactivation.

Referee 2:

1. *With regard to the comment “The authors claim as, “For the first time, the photocatalytic activity of ZIF-8 was discovered and characterized”, I would suggest to rephrase this statement.”*

Per suggestion, we have rephrased the statement in the revised manuscript (Page 11, Line 24-25)

2. *As to the comment “...antibacterial activity stems from 2-methylimidazole (H-MeIM) and zinc (Zn^{2+}) ions ...On contrary, in current study, Zn^{2+} and H-MeIM had little antibacterial effect (Fig. 2c). ... explanation their contradictory observation with reference to the given example or any other if they find in the literature. Alternatively, the authors should clearly state the different outcome and which experiments confirm their hypothesis.”*

In our work, trace leached Zn^{2+} ions of 2.28 and 2.65 $mg L^{-1}$ have indeed been detected by ICP-MS after illuminating 500 $mg L^{-1}$ ZIF-8 suspension and its mixture with *E. coli* cells (10^7 CFU mL^{-1}) over 120 min, respectively. High concentrations of Zn^{2+} ions and H-MeIM can kill a certain number of bacteria. In our experiment conditions, trace amount of Zn^{2+} ions and H-MeIM show negligible effect on bacteria reduction comparing with blank control (Zn^{2+} ions at 3 $mg L^{-1}$ gives 93.5% inactivation efficiency; H-MeIM at 7 $mg L^{-1}$ gives 73.7% efficiency; blank control: light irradiation without any catalyst gives 76.3% inactivation efficiency under the identical conditions). It is

noteworthy that the levels of Zn^{2+} released from ZIF-8 is much lower than the minimum inhibitory concentration (MIC, 31.25 mg L⁻¹) and minimum bactericidal concentration (MBC, 250 mg L⁻¹) (Fig. S7 and Table S1).

In the reference (*Adv. Mater. Interfaces.*, 2018, 5,1800167), the ZIF-8 nanocomposite exhibited antibacterial activity against *E. coli* due to the releasing of Zn^{2+} and H-MeIM. This is actually consistent with the results and discussion as mentioned above. Unfortunately, the authors did not offer inactivation efficiency and the detailed experimental data, such as the released Zn^{2+} and H-MeIM concentrations, amount of ZIF-8, initial concentration of *E. coli* cells and dilution ratio of the sample. It is thus difficult for us to make direct comparison.

Remarkably, the incorporation of ZIF-8 with light irradiation could decrease the count of bacterial colonies by six orders of magnitude, due to the high effectiveness of ROS disinfection mechanism. The antibacterial effect of light irradiated ZIF-8 (> 99.9999% inactivation efficiency) is well beyond that of ZIF-8 in dark (52.5% inactivation efficiency) and light alone (76.3% inactivation efficiency).

For clarity, we have added the above discussions and results in the revised manuscript (Page 5, Line 16-22) and the supplementary information (Page 7, Line 1-12, Fig. S7 and Table S1).

3. *Concerning the comment “For ZIF-8, the estimated band gap of 3.3 eV (this study) is different from previously published value of 4.9 eV (Angew. Chem. Int. ed., 2011, 50, 450). It would be helpful if the authors explain their different finding compared to the literature values. ...”*

According to the equation for optical band gap calculations (band gap = 1240 / wavelength), the band gap is closely related to the spectra location of absorption edge. As mentioned in response 9 to referee 1, there are two apparent absorption peaks at 227 nm and 350 nm in the UV-vis diffuse reflectance spectra. If a wavelength edge of 250 nm was employed, a value of 4.9 eV would be obtained based on the above the equation. In our work, the absorption edge should be at 375 nm due to the LMCT transition at 350 nm for effective photocatalysis. Therefore under simulated sunlight (300 nm < λ < 1100 nm), a band gap of 3.3 eV is more relevant, as also evidenced by the results from others (*J. Alloy. Compd.*, 2017, 693, 543).

4. *With regard to the suggestion “... generation of hydroxyl radical is thermodynamically unfavorable (as also supported via scavenger experiments). This is confusing as another report*

(*RSC Adv.*, 2014, 4, 54454) describes ZIF-8 photocatalysts exhibiting efficient photocatalytic activity for methylene blue degradation under UV irradiation, which was confirmed through the detection of hydroxyl radicals ($\bullet\text{OH}$) by a fluorescence method.”

In our work, experiments including antibacterial test, quenching test and ROS detection were all carried out under neutral conditions, since the bacteria are more likely to breed in such environment. H_2O could not be oxidized to $\bullet\text{OH}$ by holes due to the lower VB level of ZIF-8 (2.1 V vs NHE) than the redox potential $\text{H}_2\text{O}/\bullet\text{OH}$ (2.68 V vs NHE), as evidenced by the EPR and quenching tests in our original manuscript. To further clarify this point, we have now carefully detected $\bullet\text{OH}$ with coumarin (Cou) by fluorescent probe method (*Electrochem. Commun.*, 2000, 2, 207-210). Fluorescence spectra of generated 7-hydroxycoumarin which could emit fluorescence at 455 nm when excited at 332 nm was measured. The unchanged fluorescence intensities at 455 nm versus time, further confirmed the absence of $\bullet\text{OH}$ radicals in the photocatalytic disinfection process.

According to the reference (*RSC Adv.*, 2014, 4, 54454-54462), photogenerated $\bullet\text{OH}$ can be detected under alkaline conditions. Per request, we have also determined $\bullet\text{OH}$ in 2 mM NaOH aqueous solution with the existence of ZIF-8, by the fluorescence method using terephthalic acid (BDC) as a probe. The results indicate that ZIF-8 do favor the formation of $\bullet\text{OH}$ in alkaline aqueous solution under light irradiation, due to the more positive VB potential of ZIF-8 than $\text{OH}^-/\bullet\text{OH}$ (1.99 V vs NHE).

Fig. S15. Time-dependent fluorescence spectra of (a) generated 7-hydroxycoumarin in ZIF-8 reaction system for $\bullet\text{OH}$ detection. 7-hydroxycoumarin could emit fluorescence at 455 nm when excited at 332 nm. (b) generated 2-hydroxyterephthalic acid in ZIF-8 reaction system containing 2 mM NaOH for $\bullet\text{OH}$ detection. 2-hydroxyterephthalic acid could emit fluorescence at 425 nm when excited at 315 nm.

We have added the above discussions and results in the revised manuscript (Page 7, Line 16-18) and the supplementary information (Page 10, Line 10-13 and Fig. S15).

Again we want to thank you and the referees for the insightful comments and for taking the time to provide critical input. We believe this manuscript is improved by this process.

REVIEWERS' COMMENTS:

Reviewer #1 (Remarks to the Author):

I am generally satisfied with the responses and revisions in the manuscript.

Reviewer #2 (Remarks to the Author):

The manuscript by Ping Li et al. describes the fabrication of antibacterial and self-cleaning filtration materials.

Overall, it was a pleasure reviewing the manuscript and the authors provided convincing evidence to answer all reviewer comments in the revised version.

In my opinion the article would be very interesting to the readers of Nature Communications.

Reviewer #1 (Remarks to the Author):

I am generally satisfied with the responses and revisions in the manuscript.

Reviewer #2 (Remarks to the Author):

The manuscript by Ping Li et al. describes the fabrication of antibacterial and self-cleaning filtration materials.

Overall, it was a pleasure reviewing the manuscript and the authors provided convincing evidence to answer all reviewer comments in the revised version.

In my opinion the article would be very interesting to the readers of Nature Communications.

We are happy to see the strong support from the reviewers!